


# Predictive skill of Atmospheric Rivers in western Iberian Peninsula

Alexandre M. Ramos[1], Pedro M. Sousa[1], Emanuel Dutra[1], Ricardo M. Trigo[1,2]

[1] Instituto Dom Luiz (IDL), Faculdade de Ciências, Universidade de Lisboa, 1749-016 Lisboa, Portugal
[2]Departamento de Meteorologia, Instituto de Geociências, Universidade Federal do Rio de Janeiro, Rio de Janeiro, 21941-916, Brazil

*Correspondence to*: Alexandre M. Ramos (amramos@fc.ul.pt)

**Abstract.** It is now clear that a large fraction of extreme precipitation and flood events across Western Europe are triggered
by Atmospheric Rivers (ARs). The association between ARs and extreme precipitation days over the Iberian Peninsula has
been well documented for western river basins.

Since ARs are often associated with high impact weather, it is important to study their medium-range predictability. Here we
perform such an assessment using the ECMWF ensemble forecasts up to 15 days for events that made landfall in western
Iberian Peninsula during the winters spanning between 2012/2013 and 2015/16. IVT and precipitation from the 51 ensemble
members of the ECMWF Integrated Forecasting System (IFS) ensemble (ENS) were processed over a domain including
western Europe and contiguous North Atlantic Ocean.

Metrics concerning the ARs location, intensity and orientation were computed, in order to compare the predictive skill (for
different prediction lead times) of IVT and precipitation analyses in the IFS. We considered several regional boxes over
Western Iberia, where the presence of ARs is detected in analysis/forecasts, enabling the construction of contingency tables
and probabilistic evaluation for further objective verification of forecast accuracy. Our results indicate that the ensemble
forecasts have skill to detect upcoming ARs events, which can be particularly useful to better predict potential
hydrometeorological extremes. We also characterized how the ENS dispersion and confidence curves change with increasing
forecast lead times for each sub-domain. The probabilistic evaluation, using ROC analysis, shows that for short lead times
precipitation forecasts are more accurate than IVT forecasts, while for longer lead times this reverses (~10 days). Furthermore,
we show that this reversal occurs for shorter lead times in areas where the ARs contribution is more relevant for winter
precipitation totals (e.g. northwestern Iberia).




## 1. Introduction

Extreme precipitation events in the Iberian Peninsula are often due to anomalous vertically integrated water vapor transport (IVT) which are generally associated with an Atmospheric River (AR, Ramos et al., 2015, Eiras et al., 2016, Ramos et al.,
2018). According to the definition of the American Meteorological Society glossary of Meteorology, ARs correspond to "a long, narrow, and transient corridor of strong horizontal water vapor transport that is typically associated with a low-level jet stream ahead of the cold front of an extratropical cyclone. The water vapor in ARs is supplied by tropical and/or extratropical moisture sources and these systems frequently lead to heavy precipitation where they are forced upward—for example, by mountains or by ascent in the warm conveyor belt. Horizontal water vapor transport in the midlatitudes occurs primarily in
atmospheric rivers and is focused in the lower troposphere" (Ralph et al., 2018).

Extreme precipitation and floods in other regions of the world have been shown to be also associated with ARs, especially on the western continental coastlines of the mid-latitudes (Guan and Waliser, 2015, Gimeno et al., 2016). The preferred regions for ARs to strike are the western coast of the continents like: California (e.g. Gershunov et al., 2017, Ralph et al., 2017, Rutz et al., 2015), South Africa (e.g. Blamey et al., 2018, Ramos et al., 2019), Chile (e.g. Viale et al., 2018, Valenzuela and Garreaud,
2019) the Iberian Peninsula (Ramos et al., 2015, Eiras et al., 2016) or even Southern Australia and New Zealand (Guan and Waliser, 2015, Kingston et al., 2016). However, their climate and socio-economics impacts are significant also in other regions of the world, as western and northwestern Europe (Lavers and Villarini, 2015, 2013, Sodemman and Stohl, 2013) or even the east cost of the US (Mahoney et al., 2016, Miller et al., 2019).

AR impacts are not always hazardous, as they can be also responsible for providing beneficial water supply (e.g. Dettinger,
2013). Lavers and Villarini, 2015 show that for western Europe, the west coast of the United States, and for the central and northeastern United States, the AR contribution to the total precipitation occurring during the winter months is in the range between 30% to 50%. In addition, Ralph et al., 2019, introduced a new scale for the intensity and impacts of ARs along the west coast of the United States, where the authors showed that weak ARs are mostly beneficial, since they can enhance water supply and snowpack, while stronger ARs tend to be frequently hazardous.

Due to its importance in the hydrological cycle, in the last years, there has been an increase in the number of studies dealing with the predictive skill of the different forecast systems in terms of ARs at short and medium range forecasts, as well as seasonal to sub-seasonal scales. Regarding the short (1–3 days) and medium-range (3–14 days) forecasts, most studies are focused on the US. Among them, Nayak et al., 2014 analyzed the skill of global numerical weather prediction models to forecast atmospheric rivers over the central United States showing that, for five different numerical models, AR occurrences
are predicted quite well for short lead times, with an increase in the location errors as the lead time increases to about 168h. The authors show that, as expected, the skill of the forecast decreases with increasing lead time in both occurrence and location. On the other hand, Martin et al., 2018, examined in detail the skill of a mesoscale numerical weather prediction system (WRF) and compared it with a global numerical weather prediction model (Global Ensemble Reforecast Dataset, Hamill et al. 2013) during AR events for the western United States. It was shown that both models present similar and important errors in low-
level water vapor flux, and consequently on the magnitude of precipitation. However, it was found that that WRF (at 9 km


horizontal resolution) can add value to the forecast when compared with a global numerical weather prediction model by means of dynamical downscaling of the medium-range forecast.

Using an adjoint model framework (Errico, 1997), it was shown for short-range forecast that both low-level winds and precipitation, for ARs striking California, are most sensitive to mid- to lower-tropospheric perturbations in the initial state in

and near the ARs (Doyle et al., 2014; Reynolds et al., 2019). This kind of studies can help identify locations of greatest sensitivity in the forecast, thus helping to plan observational campaigns that probe ARs using research aircraft and dropsondes; the dropsonde observations will then be assimilated into control forecast models, such as ECMWF. There is an AR reconnaissance and data assimilation program led by the Center for Western Weather and Water Extremes at Scripps Institution of Oceanography University of California, San Diego, USA and additional information can be found here:

https://cw3e.ucsd.edu/arrecon_overview/.

Weather forecasting uses a process called ensemble forecasting to generate multiple realizations of possible future atmospheric conditions or states. This is undertaken to take into account uncertainties in the initial atmospheric state and inadequacies in the numerical model formulations. In recent years a new approach based on the IVT forecasts (Lavers et al, 2014) has revealed that the IVT may provide earlier awareness of ARs and extreme precipitation than precipitation forecasts in different regions

of the world (Lavers et al., 2016, Lavers et al., 2017, Lavers et al., 2018). The rationale behind it is to use higher IVT predictive skill (e.g. Lavers et al., 2014; 2016) and then use the ECMWF Extreme Forecast Index (EFI, Zsoter et al., 2014). The EFI assesses how extreme the ensemble forecasts are with respect to the model climate and provide values that range between −1 and 1. Regarding the Iberian Peninsula, Lavers et al., 2018, showed, using a high-density daily surface precipitation observation, for the winters of 2015/2016 and 2016/2017, the IVT EFI has slightly more skill (than the precipitation

EFI) in discriminating extreme precipitation anomalies across the western Iberian Peninsula (Portugal and northwestern Spain) from forecast day 11 onwards. The EFI for IVT became control at ECMWF in the summer of 2019.

Since the ARs are relatively narrow corridors of strong horizontal water vapor transport, its landfall position will influence the location of a possible extreme precipitation event. In the case of the Iberian Peninsula, it was shown by Ramos et al., 2015, that the occurrence (or not) of extreme precipitation days over western river basins is highly sensitive to the latitudinal location

of the AR landfall. Therefore, it is important to obtain an objective assessment of the forecast accuracy, at different lead times regarding ARs landfall position by using the IVT. This will be explored here, through a validation procedure that is based on the observational precipitation records field.

The main objective here is twofold: a) to compare the values of IVT and precipitation forecasts at different lead times during extreme ARs striking western Iberia, using ECMWF ensemble forecasts up to 15 days for winters between 2012/2013 and

2015/16; and b) to assess the skill (or accuracy) of IVT probabilistic forecasts in terms of location landfall and intensity, through a probabilistic verification procedure, thus allowing the identification of possible model biases during extreme ARs events, and to define simple metrics which may be suitable for control purposes.




## 2. Dataset

### 2.1 Forecast dataset

The ECMWF integrated forecasting system (IFS) ensemble (ENS) operational forecasts were processed for the extended winter seasons (October to March) in four winters: 2012/2013, 2013/2014, 2014/15 and 2015/16. ENS has 51 ensemble members and the two daily forecasts initialized at 00UTC and 12 UTC with a lead time of 15 days were processed. One of these is known as the control forecast and is produced with the best estimate of the initial atmospheric state. The remaining 50 members are generated by perturbating the initial conditions. The forecast data considered were daily precipitation and IVT (both direction and magnitude); the IVT was computed using the specific humidity and zonal and meridional winds between 300hPa and 1000hPa levels (e.g. Ramos et al., 2015). The operational forecasts in the four winters had several upgrades, including model and resolution updates (accessed 18 September 2019: https://www.ecmwf.int/en/forecasts/documentation-and-support/changes-ecmwf-model). A more detailed evaluation of the impact of these model changes on forecast skill would have required a detailed analysis of the past forecasts (hindcasts) of each model version, which is beyond the scope of this study.

### 2.2. Observed precipitation dataset

In order to evaluate the operational forecasts, we have used the Portuguese national network of automatic weather stations surface provided by the Portuguese Institute of Meteorology (Instituto Português do Mar e da Atmosfera, IPMA). The data include 10 minutes accumulated precipitation from around 100 automatic weather stations over mainland Portugal, which were chosen based on a combination of tests for completeness and quality. The 10 minutes precipitation were accumulated into consecutive 12h periods staring at 00UTC and 12UTC of each day.

## 3. Comparing the predictive skill of precipitation and IVT

Firstly, a Receiver Operating Characteristic (ROC, Wilks 2006) curves analysis was performed for IVT and precipitation forecasts for Portugal. Using the observed precipitation dataset presented in Section 2.2, a list of extreme precipitation events over mainland Portugal (associated with ARs) was obtained by considering days where the mean precipitation over the considered surface stations was above the 95th percentile of all the rainy days (>1mm), and simultaneously an AR was detected in the region (IVT >450kg/m/s), following the threshold found by Ramos et al. (2015) for ERA-Interim reanalysis. Forecasts of extreme IVT and precipitation, above the 95th percentile of the corresponding model climatologies, are then compared against observations, considering a "yes" forecast if a sufficient number of Ensemble members surpass that given threshold. A ROC curve is obtained by computing Hit Rates versus False Alarm Rates (Wilks, 2006) considering a different minimum fraction of Ensemble members above the 95th percentile (from 0.1 to 1). The area under the ROC curve above 0.5 denotes skilful forecasts in respect to climatology. In Fig. 1, the ROC curves and areas for forecasts at different lead times are presented. The ROC curves analysis (Fig. 1 upper panel) clearly shows that lead time is crucial for the performance of both IVT and precipitation forecasts. For both variables, the forecast skill becomes negligible beyond 10 days lead time. While this is not


unexpected, our results show that IVT can add potential value to extreme precipitation forecasts during ARs in the mid-range
time frame (5 to 10 days). As Fig. 1 middle panel shows, around day 4-5, ROC areas are higher for IVT than for precipitation.
This result is in line with a previous work by Lavers et al. (2018), which already highlighted the added potential of probabilistic
forecasts for IVT (over precipitation forecasts) in western Iberia for 2 winters over longer lead times. Furthermore, as shown
in Fig. 1 bottom panel, when considering those days with extreme precipitation associated to ARs in Portugal, the number of
ensemble members warning for a forecast above the 95th percentile is higher for IVT than for precipitation, for all lead times.
While during the days where the precipitation ROC areas are above the IVT (<5 days) which likely reflects the more False
Alarms using the IVT, it is clear that between days 5-10 the use of probabilistic IVT forecasts can give a significant added
warning value for extreme precipitation forecasts.

We will now focus our analysis on the performance of the ECMWF probabilistic forecasts for IVT over Portugal, exploring
potential systematic biases, and trying to access model behavior and accuracy metrics at different lead times, particularly those
where we have shown that IVT can provide an added value for operational forecast.

## 4. Model bias during extreme ARs landfall events

We have defined 6 regional boxes (Fig. 2) over Western Iberia, 3 of them covering the area where IPMA surface stations are
located (North Portugal, Central Portugal and South Portugal), and also extending further north/south (Sea-North, Galicia and
Sea-South), in order to define metrics for the accuracy of the location of ARs landfall, including "hits" and "misses" in
forecasts.

### 4.1. An exemplificative case-study (January 4 2016)

To check the model performance for IVT forecast over the domain during AR events, we considered only cases where the
control analysis (+00 hours forecast) exceeded the 450kg/m/s threshold for IVT (as defined in Ramos, 2015 - where the ERA-
Interim dataset was used). These cases were considered as the "benchmark" for model forecast verification, being performed
for 00UTC and 12UTC analysis during the 4 winters spanning 2012-2016. Afterwards, forecasts up to 14 days in advance
from the control and ensemble members where compared against the analyses, through the computation of the following
metrics:

1)    Landfall distance: the distance (in km) between the location of the maximum IVT in the analysis and the forecast;

2)    Landfall IVT error: the difference (forecast minus analysis) between the IVT (in kg/m/s) at the correct location of the
landfall;

3)    AR-axis IVT error: the difference (forecast minus analysis) between the IVT (in kg/m/s) at the specific locations of
the landfall in the analysis and forecast;

4)    AR-axis angle error: the difference (forecast minus analysis) between the incidence angle (in º, respective to W→E)
at the specific locations of the landfall in the analysis and forecast.



The relevance of these metrics can be easily understood looking at a case study. In Fig.3 (upper panel, analysis), an example of an AR affecting the North Portugal box is presented. The 14 small panels show how the control forecast changed with increasing forecast daily lead time. While at short lead times the location, intensity and angle are quite similar to the analysis, at longer lead times the control forecast becomes less accurate, and some of them predicting an IVT magnitude, axis orientation and landfall totally disconnected from reality. To better estimate the possible future weather, ensemble forecasts are generated, as discussed below.

This evolution of the forecast is well depicted in Fig. 4, where the metrics for each lead time are summarized. As lead time increases, it is notable how the AR was being predicted further north (blue solid line) than where it was observed. In fact, for lead times of -9 days, -11 days and -14 days, there was no predictive skill of AR affecting the defined boxes (either forecasted much further north, or not forecasted at all, as depicted by the open circles), and in accordance with the corresponding subplot observed in Fig. 3. Consequently, as the landfall distance error increases, the IVT error at that location also increases (red solid line in Fig. 4). When considering the AR-axis IVT error, the decrease with longer lead times is smaller, showing that despite the error in the actual position of the AR, its maximum intensity was well forecasted for most of the period. Regarding the angle of incidence of the AR, it was mostly zonal, both in the analysis and most forecasts, thus with small error. Nevertheless, and as seen in Fig. 3, during mid-range lead times (around 7-10 days) forecasts tended to tilt the AR NW→SE over Western Iberia, as depicted by the arrows in Fig. 4. Again, from this point forward we will focus ensemble forecast analysis.

Once again taking the case study occurred on January 4 2016 (shown in Fig. 3 as an example), we computed the same metrics described above also for each member of the 50-strong ensemble of the IFS Probabilistic forecast and present these results in Fig. 5. The errors for the control and the ensemble mean are presented for landfall distances and IVT errors (Fig. 5), as well as the spread in the ensemble forecast, shown here as the 25th and 75th percentiles of the ensemble computed metrics distribution. In this case, the control forecast errors and the ensemble mean are in good agreement with the dispersion of the ensemble forecast, increasing both as expected with the lead times. Regarding the landfall distance, it was found that the error increase with the lead time and in this case the control forecast error at lead time -12 days is higher than the ensemble mean and even the ensemble dispersion. The 50 members of the ensemble for lead time -5 day (Fig. S1) and for -12 days (Fig. S2) are shown in supplementary material Fig. S1 where it can be compared with the control IVT forecast shown in Fig. 3. One can see, that the ensemble members for the shorter lead time are in better agreement and closer to reality. When looking to -12 days lead time (Fig. S2), the dispersion of AR location is much higher when compared with the realty and even with the control forecast shown in Fig. 3. In addition, both Landfall IVT error and AR-axis error (Fig. 5), for both control and ensemble members, show a decrease in the forecasted IVT as we consider longer lead times.

### 4.2. Mean performance of the ECMWF forecasts during 2012-2016

In the previous section we have analyzed one case study, for illustrative purposes how, evaluating the forecast for both the control and ensemble members when compared against the analyses. However, it is now necessary to compute the mean performance and metrics of the IVT forecasts for all events occurring during the extended winters of the study period (from




2012-2013 to 2015-2016) for the control simulation and for the ensemble forecast. Accordingly, similarly to the case study presented in Fig. 3 to Fig. 5, the same metrics have been computed for all AR landfall cases (207), and the average error is obtained as presented for the control forecast in Fig.6 and for the ensemble forecast in Fig. 7.

When analyzing the control ECWMF forecast of ARs over Western Iberia (Fig .6), it is possible to observe some systematic errors/biases in the IFS. Regarding the mean errors (Fig. 6) a northward landfall bias is systematically found for longer lead times, especially for those longer than 5 days, which can reach up to 800km at +14 days. In addition, regarding the AR-axis angle error, the results show a slightly negative bias in respect to those actually observed. Since we consider the 0º angle as west-east and that AR events over Portugal tend to present a southwest-northeast orientation (Ramos et al., 2015), this

systematic bias reflects a lower tilt in the AR orientation at longer time forecasts, or in other words, a tendency for more zonal forecasts (Fig 6). When analyzing the IVT magnitude, both landfall IVT error and AR-axis IVT error have a negative bias, as a result of: i) the error in the landfall location; ii) an underestimation of the ARs intensity. Comparing both metrics in more detail, it can be noted that the ARs-axis IVT bias is lower than the landfall IVT error, showing that while the IFS is forecasting the intensity of the ARs quite well (with just a small underestimation in intensity associated to lead times), the landfall location

bias leads to significant IVT forecast errors at the location where the AR actual landfall is observed. The mean absolute errors were also computed for the same metrics (Fig. 6), unsurprisingly presenting higher errors for landfall distance, location biases occur both northwards and southwards, thus partially canceling themselves, as shown in Fig. 6. Nevertheless, this difference is not that large, thus reinforcing the systematic tendency for a bias towards the north in longer lead time forecasts. A similar rationale is applicable to the incidence angle, where errors in the tilt of the ARs in different directions tend to cancel out. As

so, in terms of absolute errors, mean absolute errors tend to be around 45º at longer lead times (Fig. 6). Overall it is possible to affirm that the case study evaluated in Fig. 5 presents similar biases to those obtain here with the average of the entire set of ARs considered.

The mean weighted errors/biases for probabilistic forecasts (i.e. all the 50 ensemble members) for all events are presented in Fig. 7. The same methodology as in Fig. 5 and Fig. 6 is followed here, and the ensemble mean is presented, along with the

spread in the ensemble forecast (shown here as the 25th and 75th percentiles forecast distribution). The results are very similar to the ones found for the control forecast (Fig. 6) with a positive bias (northly) in the position of the AR landfall as we move to higher lead times, along with a negative bias (less AR intensity) in the landfall IVT error and AR -axis IVT error. However, the dispersion in the ensemble forecast is higher in the landfall distance than the in the other IVT error metrics, reinforcing that the model forecasts the ARs but lacks skill in forecasting the right location of their landfalls.

It is vital to stress the use of the ensemble forecast in this kind of research. In the example shown in section 4.1, we only using the control forecast (which is produced with the best available data and unperturbed model) for illustrative purposes on how the metrics errors were computed (Fig. 3 and Fig. 4). However, in Fig. 6 and Fig. 7., we compare the control forecast with the ensemble mean and the ensemble errors metrics. Both control forecast and ensemble forecast Landfall and IVT error shows a northly bias and a negative IVT bias on the landfall location respectively. Ramos et al., 2016, showed that the number of ARs

is relatively lower when compared with the contiguous western France or even the UK using the ERA-Interim reanalysis. This





mean that on average most of the AR go north, and the ones hitting Iberia are not that frequent. Taking this into account, one can hypothesize, that the northerly bias and respective negative IVT intensity in the ARs at longer lead times, can be that the model tends to its climate, which is to have AR further North as shown in Ramos et al., 2016. A similar behaviour occurs for the blocking frequency (Euro-Atlantic sector and the Pacific sector) using the the NCEP Climate Forecast System version 2

(CFSv2), in which for longer lead time the model tends to reach its climatology (Jia et al., 2014).

## 5. Objective verification of the IVT forecasts

In the previous section we proposed some metrics to analyze control and probabilistic IVT/ARs forecasts errors in the IFS for the Iberian Peninsula. In this section, we provide an objective verification of the IVT forecast, but considering in greater detail

the landfall location and how to use it for a possible application in terms of control forecast for ARs landfall. Firstly, and bearing in mind the regional boxes presented in Fig. 2, and the case study presented in Fig. 3, the percentage of ensemble members providing correct/incorrect forecasts regarding the regional boxes is summarized in Fig. 8 upper panel. As it can be seen, the forecast issued the day before the event was almost perfect, with most members predicting the location correctly in the North Portugal box (green bar). As lead time increases, the percentage of correct forecast decreases until day 5, but still a

large fraction of members predicts the AR to make landfall in one of the adjacent boxes (yellow bars), until around day 7. At longer lead times, the percentage of members predicting landfall in boxes further away (red bars) and predicting no landfall or no AR (brown bar) increases significantly.

As done in section 4, we now present the contingency table for the ensemble forecasts of IVT (events >450 kg/m/s) for all the events during the 4 winters considered in this study (Fig. 8, lower panel). Each different box corresponds to a lead time (from

1 to 14 days) and the different boxes corresponds to the observed vs predicted landfall location corresponding the bluish colors correspond to the location of each landfall box shown on Fig. 2. The box for lead time 1 day presents additional info to help reading the contingency tables: i) the x-axis representing "yes" forecasts for each box; ii) the y-axis representing "yes" observations for each box; iii) the color code presented for day +1 corresponds to the boxes presented in Fig. 2); iv) the last column and row, with the white circle and crosses, represent events that have been observed but not predicted and vice-versa,

respectively. Note that the numbers in the left axis represent the number of observed events in each box. A perfect forecasting model would only present values in the diagonal (NxN).

Results confirm what was partly shown in Fig. 7, as the error in the landfall locations increases with the lead time, and an increasing fraction of the ensemble members forecasting the landfall outside the Iberian Peninsula (further north of Galicia or further south of Algarve) or not even forecasting an AR. However, for shorter lead times (day -1 to day-3) the forecast error is

quite low, with the ARs landfall being predicted very well, considering the small size of the regional boxes (less than 1º latitude each). In addition, the contingency table also confirms the northward landfall bias of most forecasts, with the left side of the table being more populated, meaning that forecasted location is more frequent in the northern boxes, when compared to observations. It is also shown that a few ensemble members pick up the AR, therefore it can be argued that system IFS is skillful, however with low probability of occurrence in the right location.



Finally, different verification forecast metrics widely used are also computed for the different ARs landfall cases for each box (Fig. 2). The metrics used are the probability of detection (POD), Success Rate (RS), False Alarm Rate (FAR) and the BIAS (Wilks, 2006) and their formulation is shown in the supplementary material. As mentioned before, and due to the increase in landfall error with lead time, these systematic errors are also expected to be reflected in the verification forecast metrics. Both POD and RS decrease as we move further ahead in lead times, getting closer to zero from lead time higher than 5 days (Fig.

9). On the contrary, the FAR is expected to increase with the lead time, staying above 0.5 in all boxes from lead times of 5 days or more. The relatively fast decline with lead time in these metrics is not surprising, as they are computed for very small target areas, and just reflects that a very accurate forecast of landfall location becomes difficult at lead times longer than 5 days, when considering mesoscales. Still, as shown by the metrics from Fig.8, and in the ROC curves analysis shown in Section 3, at a synoptic scale, the model is able to forecast high probabilities of an AR affecting Western Iberia at longer lead times.

This suggests that an effective warning system can be developed with reasonable lead times, although very detailed local forecasts of AR activity can only be achieved at short time scales.

## 6. Conclusions

The occurrence (or not) of extreme precipitation days in different river basins is highly sensitive to the latitudinal location of

the ARs landfall as shown for the Iberian Peninsula (Ramos eta l., 2015). This is due the ARs being relatively narrow corridors of strong horizontal water vapor transport, therefore their landfall position has influence on the occurrence of a possible extreme precipitation event and its specific location. With this in mind, we assessed the forecast accuracy at different lead times regarding ARs landfall position, intensity and incidence angle by using the IVT. To achieve this goal, we used ECMWF ensemble forecasts up to 15 days, for extended winter seasons between the winters of 2012/2013 and 2015/2016, and assessed

the skill (or accuracy) of IVT probabilistic forecasts through a probabilistic verification procedure.

The main conclusions are as follows:

•	The IVT forecasts shows higher predictive skill than precipitation forecasts for lead times higher than 5 days, when considering extreme precipitation events associated to ARs over Portugal. In addition, we show that there is a higher agreement amongst the IVT ensemble members for early awareness at such lead times, than that found for the precipitation ensemble.

•	We identified the systematic errors in AR forecasts using a designed objective verification scheme for IVT/ARs applied to the ECMWF Ensemble. There is a good capability predictive skill of the model in terms ARs landfall over the domain at short term forecast. However, at longer lead times, the location of the landfall is less reliably, and ARs tend to be predicted too much to the north in western Iberian Peninsula, and their intensity tends to be underestimated.

•	In addition, when using the ensemble members to check the forecast skill for the specific ARs landfall location (using

6 regional boxes), it becomes clear that the predictive skill at larger spatial scales (the entire domain) tends to be reasonable, while the predictive skill at regional scales tends to be considerably smaller.

•	These results show the potential added value to forecast mid-range AR related precipitation events using the IVT, and to further develop warning systems based on a very simple methodology.





Finally, we presented a methodology that can be used in a control context, consisting on the probabilities of ARs striking
different regional boxes. This probability is based on the fraction of Ensemble members providing IVT forecasts above a
threshold in each box, thus providing an estimate on the probabilities of occurrence and on the expected landfall location. As
our analysis for increasing lead times shows, confidence on detailed control forecasts should quickly rise at lead times shorter
than a week, but early awareness can be expected at longer lead times. This methodology can be easily replicated using different
forecast systems (e.g. the Global Forecast System, GFS) and applied to different regions of the globe after a similar verification
as we proposed is performed.

**Author contribution**

A.M.R., P.M.S., E.D. conceived and designed the experiments, P.M.S and E.D. performed the computations and A.M.R.,
P.M.S., E.D., R.M.T, analysed the data and wrote the paper.

**Acknowledgments**

A.M.R., P.M.S. and R.M.T. were supported by the project Improving Drought and Flood Early Warning, Forecasting and
Mitigation using realtime hydroclimatic indicators (IMDROFLOOD) funded by Fundação para a Ciência e a Tecnologia,
Portugal (FCT, Portugal) (WaterJPI/0004/2014). A.M.R. was also supported by the Scientific Employment Stimulus 2017
from FCT (CEECIND/00027/2017). The authors would like to thank David Lavers for the helpful comments on this
manuscript.






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



**Figure captions**


Figure 1. Receiver Operating Characteristic curves (ROC curves) for the IVT and precipitation ensemble forecasts during Atmospheric River days (ARs) from the ECMWF model, using Portuguese surface meteorological stations during the 2012-2016 extended winters (October-March) as a benchmark, and considering events above the 95th percentile. Solid lines are for the IVT and dashed lines for precipitation. Different curve colors represent different lead times for the forecasts (1, 5, 9 and

13 days) (upper panel). Area under the ROC curves presented in a), for lead times up to 14 days (middle panel). Mean percentage of ensemble members forecasting IVT (pink) and precipitation (purple) above the 95th percentile for lead times up to 14 days during extreme rainfall events associated to ARs (observed precipitation above the 95th percentile associated to an AR over Western Iberia) (lower panel).

Figure 2. The six regional boxes considered for the verification of IVT probabilistic forecasts in Western Iberia at lead times up to 14 days: i) sea North; ii) Galicia; iii) North Portugal; iv) Central Portugal; v) South Portugal; vi) sea South.

Figure 3. Example of the evolution of the Operational Forecast of the IVT in an event affecting Western Iberia. a) Analysis of the IVT fields on January 4 2016 at 12UTC. In addition, operational forecasts for that date at different lead times, from 1 to

14 days.

Figure 4.  Example of the evolution of the accuracy of the Operational Forecast of the IVT in the event presented in Figure 3. Solid blue line represents the error in the location of the maximum IVT between observation and each forecast (in km). The solid red line shows the error in the IVT (Kg/m/s) for each forecast at the real landfall location (where the maximum IVT was

observed), while the red dashed curve represents the error in the maximum IVT between the observed and each lead time forecast, independently of the location in each forecast. Black arrows represent the errors in the angle (in degrees) of the AR axis for each forecast. Blank circles represent forecasts where the maximum IVT did not surpass a minimum threshold of 450 Kg/m/s.

Figure 5. Summary of the evolution of the accuracy of the Ensemble Forecast of the IVT in the event presented in Figure 3. Landfall distance errors (in km) for the Operational forecast (thin black line), the mean of the Ensemble Forecast (thick solid colored line) and the spread of the Ensemble (shading). In addition, the IVT error (in Kg/m/s) at the location of observed landfall and the location of the maximum IVT in each forecast are also shown.






Figure 6. Statistics for the verification of the accuracy of the Operational Forecast of IVT for all events affecting Western Iberia during the extended winters between 2012 and 2016. Upper panel is relative to the mean errors, while the lower panel correspond to the absolute errors. Solid blue line represents the error in the location of the maximum IVT between observation and each forecast (in km). The solid red line shows the error in the IVT (Kg/m/s) for each forecast at the real landfall location (where the maximum IVT was observed), while the red dashed curve represents the error in the maximum IVT between the observed and each lead time forecast, independently of the location in each forecast. Black arrows represent the errors in the angle (in degrees) of the AR axis.

Figure 7. Statistics for the verification of the accuracy of the Ensemble Forecast of IVT for all events affecting Western Iberia during the extended winters between 2012 and 2016. a) mean Landfall distance errors (in km) for the Operational forecast (thin black line), the mean of the Ensemble Forecast (thick solid colored line) and the spread of the Ensemble (shading). b) As in a), but for the mean IVT error (in Kg/m/s) at the location of observed landfall. c) As in b), but at the location of the maximum IVT in each forecast.

Figure 8. Percentage of Ensemble members forecasting IVT above 450 Kg/m/s in each of the regional boxes for each lead time. Green bars represent an accurate forecast, in the box where the maximum IVT was observed. Yellow bars represent a forecast in an adjacent box to where it was actually observed. Red bars represent a forecast in one of the remainder boxes. The bars in the last line represent a completely missed forecast, by either: i) no AR forecast; ii) AR forecast outside of the 6 considered boxes in Western Iberia. (upper panel). Contingency tables for the accuracy of IVT forecasts using the ECMWF Ensemble, from lead times ranging between 1 and 14 days, for all days during the extended winters spanning 2012-2016. The red shading represents the percentage of observations versus forecasts. Note that a perfect forecast system would only present shadings in the diagonal, as the y-axis represents observed events in each box box (as presented in Figure 2) and the x-axis represents forecasts in each box. The number of events in each box is shown in by the blue arrow. The last row/column represent either: i) observations/forecasts outside of the 6 considered boxes; ii) no AR observed/predicted (lower panels)

Figure 9. Forecast verification metrics for IVT exceedances (>450 Kg/m/s) using the ECMWF Ensemble forecast system during the 2012-2016 extended winters in Western Iberia, and for lead times between 1 and 14 days. Coloured bars represent metrics for individual regional boxes, as depicted in Figure 2. Solid lines and circles represent the corresponding metrics but considering the entire domain where the regional boxes are located.

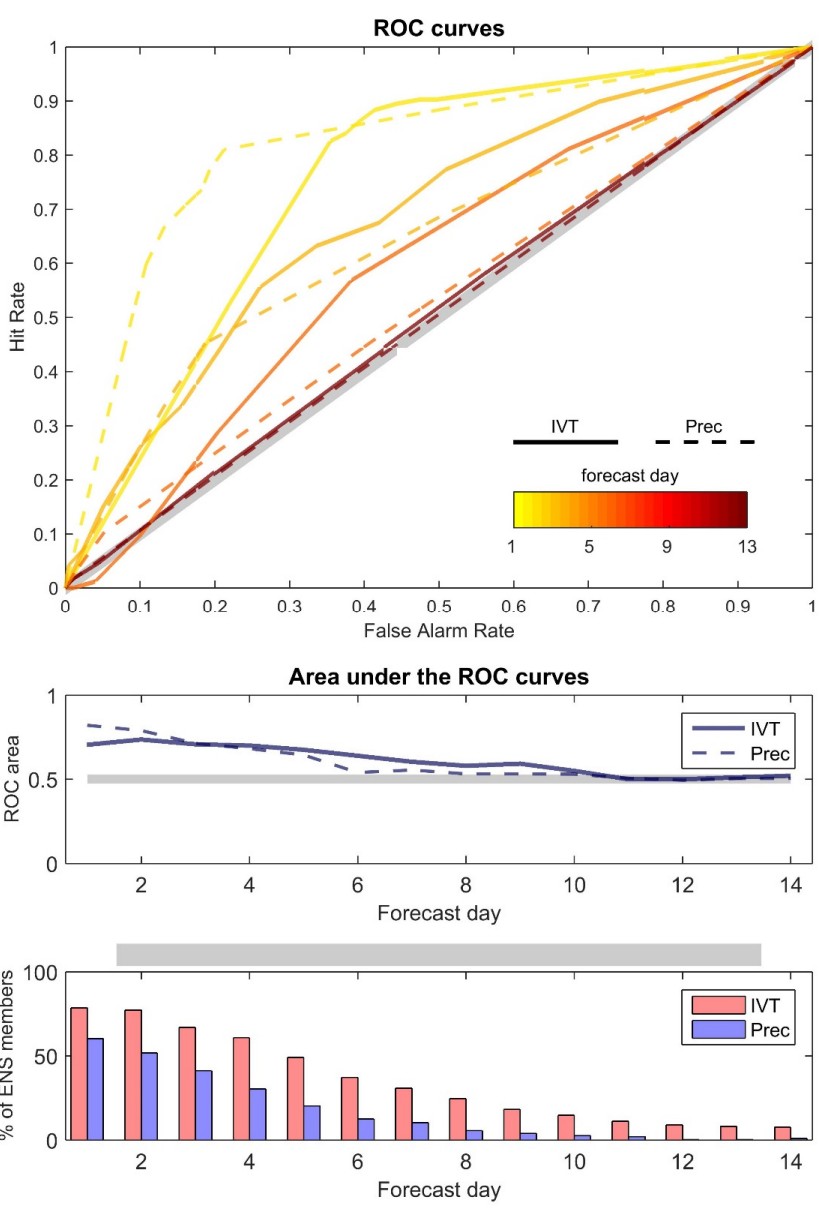

**Figure 1.** Receiver Operating Characteristic curves (ROC curves) for the IVT and precipitation ensemble forecasts during Atmospheric River days (ARs) from the ECMWF model, using Portuguese surface meteorological stations during the 2012-2016 extended winters (October-March) as a benchmark, and considering events above the 95[th] percentile. Solid lines are for the IVT and dashed lines for precipitation. Different curve colors represent different lead times for the forecasts (1, 5, 9 and 13 days) (upper panel). Area under the ROC curves presented in a), for lead times up to 14 days (middle panel). Mean percentage of ensemble members forecasting IVT (pink) and precipitation (purple) above the 95[th] percentile for lead times up to 14 days during extreme rainfall events associated to ARs (observed precipitation above the 95[th] percentile associated to an AR over Western Iberia) (lower panel).

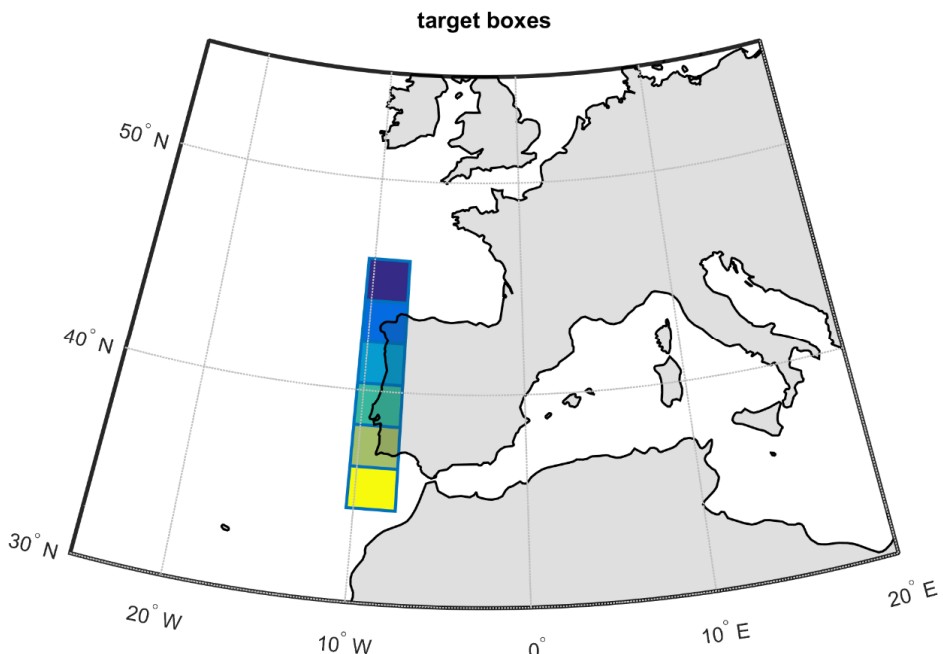

**Figure 2.** The six regional boxes considered for the verification of IVT probabilistic forecasts in Western Iberia at lead times up to 14 days: i) sea North; ii) Galicia; iii) North Portugal; iv) Central Portugal; v) South Portugal; vi) sea South.



**Figure 3.** Example of the evolution of the Operational Forecast of the IVT in an event affecting Western Iberia. a) Analysis of the IVT fields on January 4 2016 at 12UTC. In addition, operational forecasts for that date at different lead times, from 1 to 14 days.

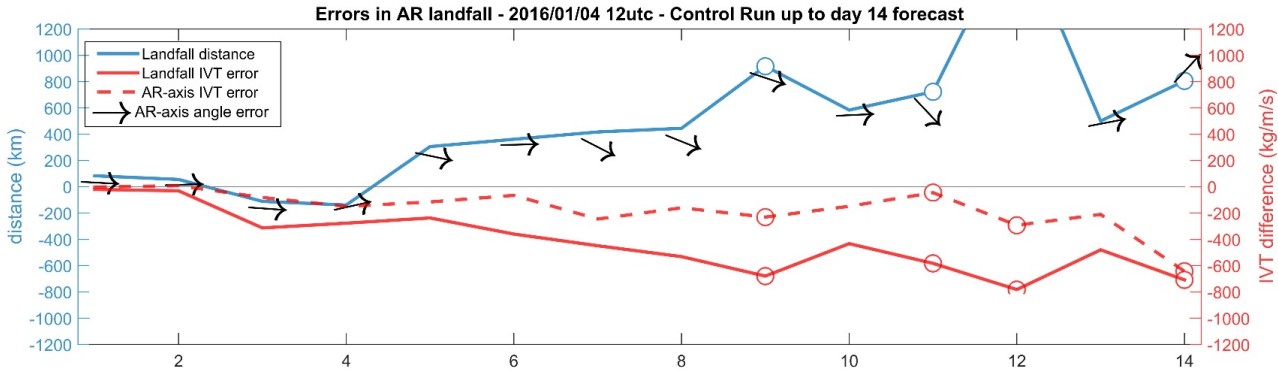

**Figure 4.** Example of the evolution of the accuracy of the Operational Forecast of the IVT in the event presented in Figure 3. Solid blue line represents the error in the location of the maximum IVT between observation and each forecast (in km). The solid red line shows the error in the IVT (Kg/m/s) for each forecast at the real landfall location (where the maximum IVT was observed), while the red dashed curve represents the error in the maximum IVT between the observed and each lead time forecast, independently of the location in each forecast. Black arrows represent the errors in the angle (in degrees) of the AR axis for each forecast. Blank circles represent forecasts where the maximum IVT did not surpass a minimum threshold of 450 Kg/m/s.

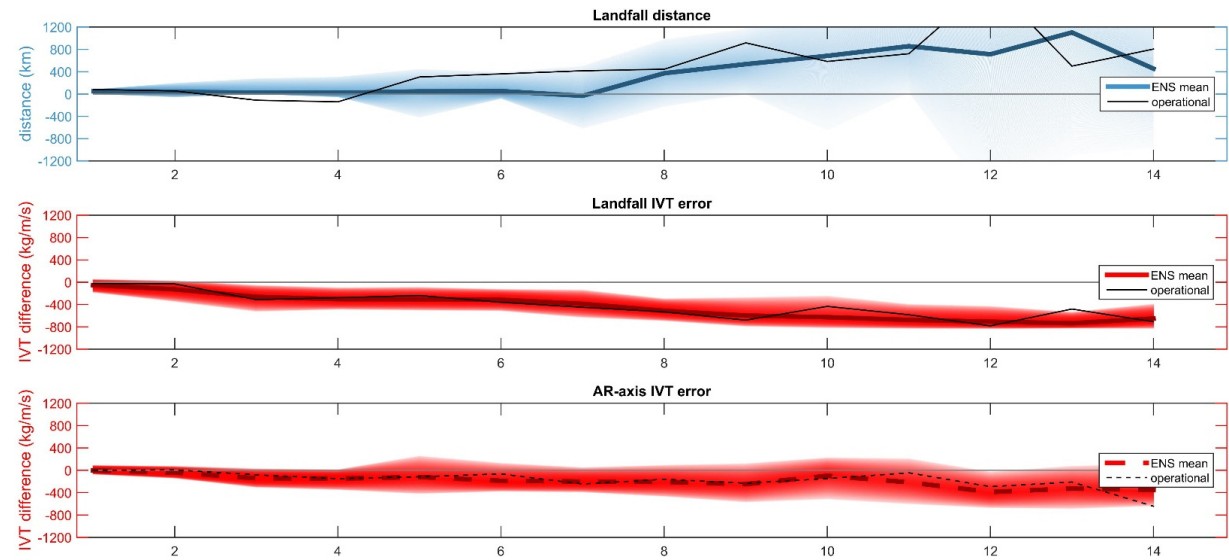

**Figure 5 .** Summary of the evolution of the accuracy of the Ensemble Forecast of the IVT in the event presented in Figure 3. Landfall distance errors (in km) for the Operational forecast (thin black line), the mean of the Ensemble Forecast (thick solid colored line) and the spread of the Ensemble (shading). In addition, the IVT error (in Kg/m/s) at the location of observed landfall and the location of the maximum IVT in each forecast are also shown.

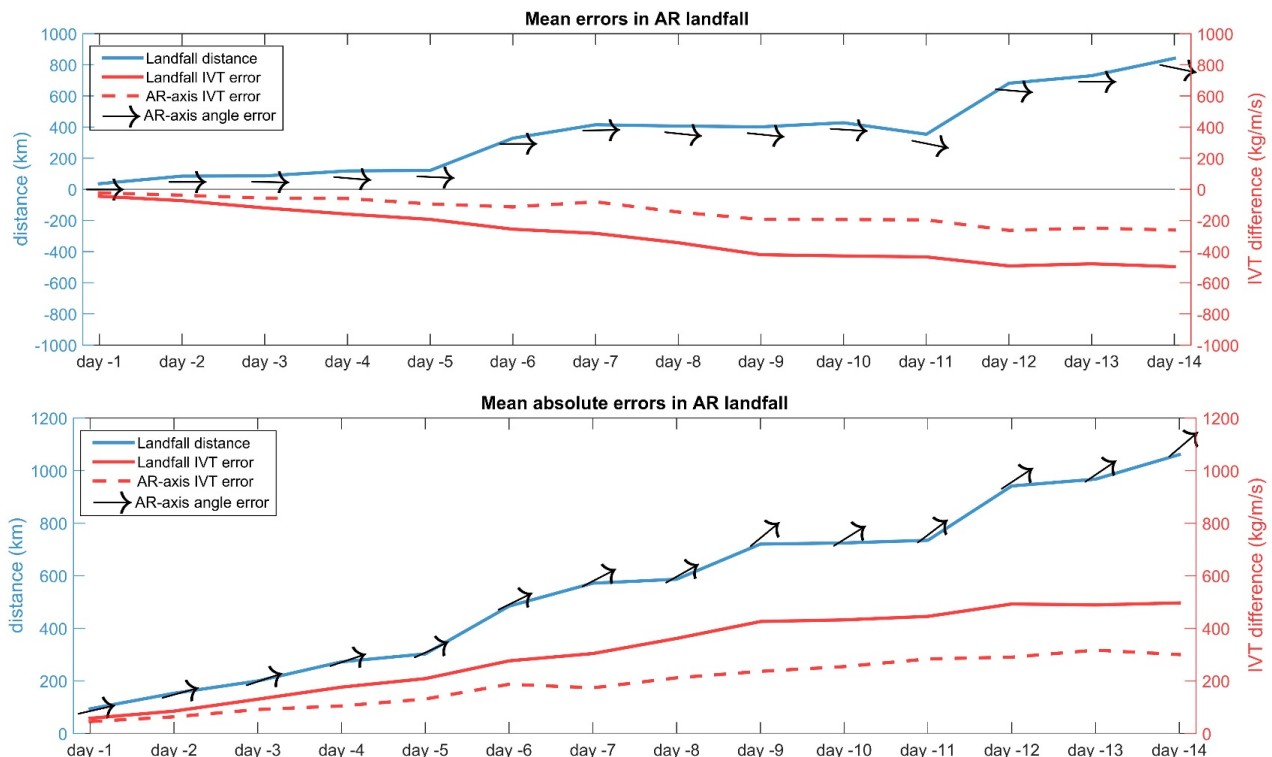

**Figure 6.** Statistics for the verification of the accuracy of the Operational Forecast of IVT for all events affecting Western Iberia during the extended winters between 2012 and 2016. Upper panel is relative to the mean errors, while the lower panel correspond to the absolute errors. Solid blue line represents the error in the location of the maximum IVT between observation and each forecast (in km). The solid red line shows the error in the IVT (Kg/m/s) for each forecast at the real landfall location (where the maximum IVT was observed), while the red dashed curve represents the error in the maximum IVT between the observed and each lead time forecast, independently of the location in each forecast. Black arrows represent the errors in the angle (in degrees) of the AR axis.

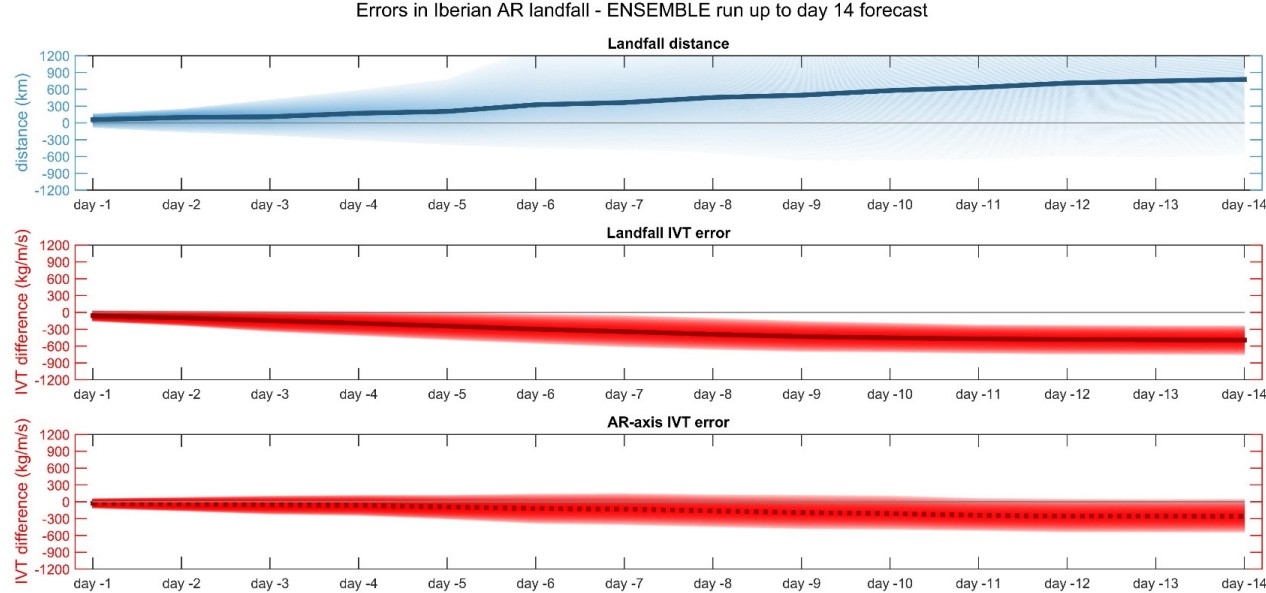

**Figure 7** – Statistics for the verification of the accuracy of the Ensemble Forecast of IVT for all events affecting Western Iberia during the extended winters between 2012 and 2016. a) mean Landfall distance errors (in km) for the Operational forecast (thin black line), the mean of the Ensemble Forecast (thick solid colored line) and the spread of the Ensemble (shading). b) As in a), but for the mean IVT error (in Kg/m/s) at the location of observed landfall. c) As in b), but at the location of the maximum IVT in each forecast.

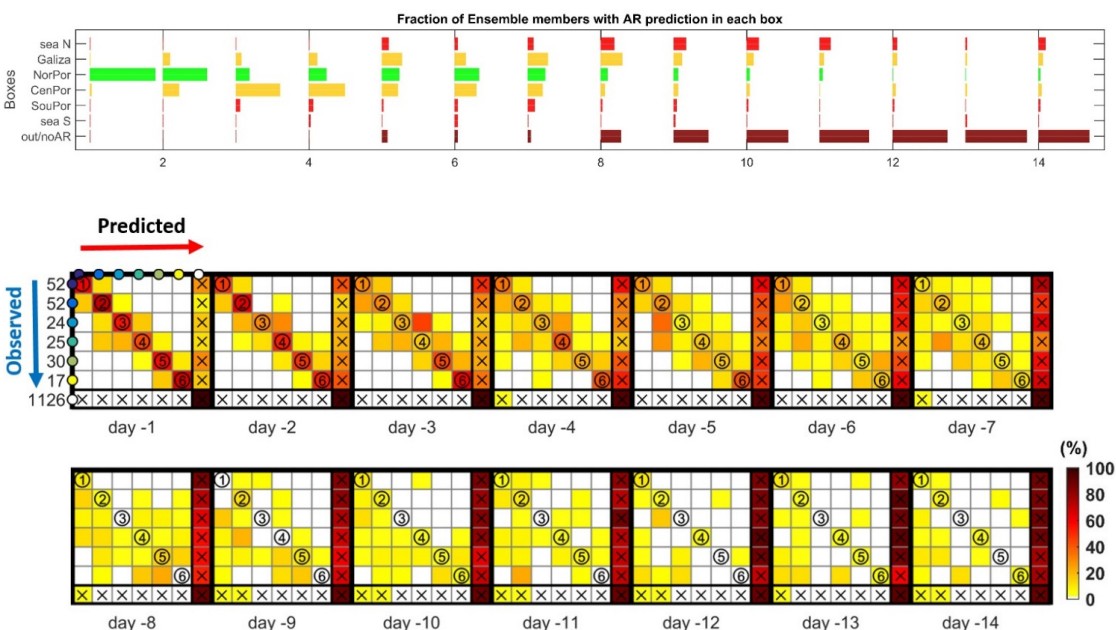

**Figure 8 –** Percentage of Ensemble members forecasting IVT above 450 Kg/m/s in each of the regional boxes for each lead time. Green bars represent an accurate forecast, in the box where the maximum IVT was observed. Yellow bars represent a forecast in an adjacent box to where it was actually observed. Red bars represent a forecast in one of the remainder boxes. The bars in the last line represent a completely missed forecast, by either: i) no AR forecast; ii) AR forecast outside of the 6 considered boxes in Western Iberia. (upper panel)

Contingency tables for the accuracy of IVT forecasts using the ECMWF Ensemble, from lead times ranging between 1 and 14 days, for all days during the extended winters spanning 2012-2016. The red shading represents the percentage of observations versus forecasts. Note that a perfect forecast system would only present shadings in the diagonal, as the y-axis represents observed events in each box box (as presented in Figure 2) and the x-axis represents forecasts in each box. The number of events in each box is shown in by the blue arrow. The last row/column represent either: i) observations/forecasts outside of the 6 considered boxes; ii) no AR observed/predicted (lower panels)


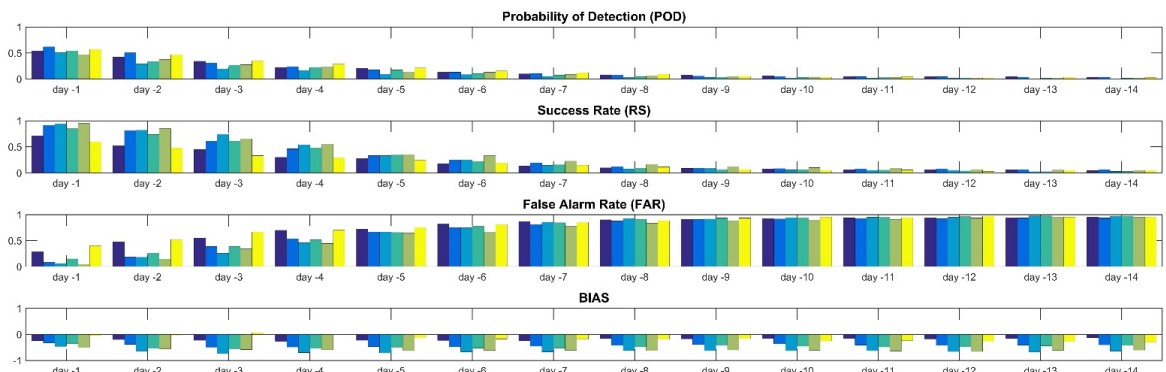

**Figure 9.** Forecast verification metrics for IVT exceedances (>450 Kg/m/s) using the ECMWF Ensemble forecast system during the 2012-2016 extended winters in Western Iberia, and for lead times between 1 and 14 days. Coloured bars represent metrics for individual regional boxes, as depicted in Figure 2. Solid lines and circles represent the corresponding metrics but considering the entire domain where the regional boxes are located.