# Peer review of "Predictive skill of Atmospheric Rivers in western Iberian Peninsula"

_Natural Hazards and Earth System Sciences, 2019_

## Referee Comment (RC1) · Maximiliano Viale (Referee) · 17 Oct 2019

General comment This paper evaluates the ECMWF ensemble forecasts up to 15 days for AR that make landfall on western Iberian Peninsula. The paper is straightforward, reads well, their results have important relevancies for weather forecasting in the region. My criticism is minimal and relates to the presentation in some parts of the manuscript which may need to be improved. Overall, this article is welcome to the weather forecasting and atmospheric community in the region, and my recommendation is to publish this article in Nat. Hazard journal after considering some minor comments provided below to improve the presentation.

Minor Comments • Line 53: Replace "cost" by "coast" • In the section 3, the comparison of the model output forecasts against the observations were done considering separately the sites or point with observations or using a regional average with observation? Please explain a little bit more about this point. • In line 132: what period does it correspond to the model climatologies? Please specify. • In line 133: what does it mean a sufficient number of ensemble members? Please specify. • In Fig 3 may be is not necessary adding all subpannels with all the days. Perhaps the authors can incorporate subpanels only every 3 days would be sufficient to show the idea and not overcharge the figure. These are very small and hard to visualize. • Fig 8 has too much information and of the different type. Considering splitting into two figures: the upper (percentages) and lower (contingency tables) panels. In caption indicate that the percentages correspond to the case study shown in Fig 3. The shading color codes of bars in contingency tables could be discrete (using the 5 subdivision) rather than continuous to better visualize the percentages. The first sentence in caption for these contingency tables could be rewritten as follow: Contingency tables for the accuracy of AR-related IVT forecasts by the ECMWF ensemble system, for lead times ranging between 1 and 15 days during winters spanning 2012-2016.

---

## Referee Comment (RC2) · Anonymous Referee #2 · 19 Nov 2019

Please see the attached pdf.

Please also note the supplement to this comment:
https://www.nat-hazards-earth-syst-sci-discuss.net/nhess-2019-317/nhess-2019-317-RC2-supplement.pdf
* * *

---

## Author Comment (AC2) · 5 Feb 2020

Please see Supplement pdf.

Please also note the supplement to this comment:
https://www.nat-hazards-earth-syst-sci-discuss.net/nhess-2019-317/nhess-2019-317-AC2-supplement.pdf
* * *

---

## Author Response (AR1)

The authors would like to thank both reviewers for their useful comments which helped to improve the manuscript.

In addition to the changes asked by the reviewers, we have made improvements also in Figure 1. The ROC curves (Figure 1a) and Areas under the ROC curves (Figure 1b) where re-computed, using a bootstrap process (with 1000 repetitions). The envelopes presented in Figure 1b depict confidence intervals and correspond to the $10^{th}$ and $90^{th}$ percentiles.

Considering that Figure 8 was split into two news figures (New Figure 7 and 8) and to keep a manageable number of figures, the authors combined in Figure 4 both previous Figures 4 and 5 (case study) without losing its readability.

The revised figures and corresponding figure captions are shown below after the reply to the reviewer comments.

In addition, the manuscript with track changes is also included at the end.

**Reviewer #1. Maximiliano Viale**

**General comment**
This paper evaluates the ECMWF ensemble forecasts up to 15 days for AR that make landfall on western Iberian Peninsula. The paper is straightforward, reads well, their results have important relevancies for weather forecasting in the region. My criticism is minimal and relates to the presentation in some parts of the manuscript which may need to be improved. Overall, this article is welcome to the weather forecasting and atmospheric community in the region, and my recommendation is to publish this article in Nat. Hazard journal after considering some minor comments provided below to improve the presentation.

**Minor Comments**

Line 53: Replace "cost" by "coast"
The typo was corrected.

In the section 3, the comparison of the model output forecasts against the observations were done considering separately the sites or point with observations or using a regional average with observation? Please explain a little bit more about this point.
In section 3, we considered the precipitation averaged in all the observed station precipitation dataset in Portugal to define "yes" or "no" extreme precipitation observations. These "yes/no" extreme precipitation events are compared against forecasts for precipitation and IVT. These correspond to the outputs from the forecast model within the same domain over Portugal for both variables, and a forecast is considered as an extreme one if it exceeds the $95^{th}$ percentile

Line 132: what period does it correspond to the model climatologies? Please specify.
For all series considered as "extreme" (both forecast and observations) in the ROC curves analysis, we defined thresholds based on the 95th percentiles, and using the longest period available for the

dataset. This is needed to ensure that we obtain thresholds which are representative for the specific realm of each independent dataset (station data VS model data), which obviously have different natures and magnitudes/ranges. Thus, they need to be compared using percentiles, and not absolute values. In the case of the Operational/Ensemble forecasts, we considered the period of available data, i.e. winters between 2011-2012 and 2015-2016. For each forecast day we defined the specific percentiles in the target domain using this period. We acknowledge that, in this context, the use of the word "climatology" might be abusive, using this rather short period. In this sense, this information was included in the new version of the manuscript (see the answer to the following query).

Line 133: what does it mean a sufficient number of ensemble members? Please specify.
The minimum fraction of ensemble members presenting a "yes forecast" varies between 0.1 and 1. So 0.1 means that 10% of the ensemble members have a "yes forecast" while 1 correspond to the totality of the ensemble members. This is how the ROC curves are computed: Hit Rates and False Alarm Rates are calculated repeatedly for each one of these varying thresholds (of minimum ensemble members presenting a "yes forecast"), thus enabling the computation of the data presented in Figure 1.

Considering the last three comments of the reviewer, the first paragraph has been revised as follows:

*Firstly, a Receiver Operating Characteristic (ROC, Wilks 2006) curve analysis was performed for IVT and precipitation forecasts for mainland Portugal. To begin with, using the observed precipitation dataset presented in Section 2.2, the mean precipitation (averaged over all mainland Portuguese stations) was computed. Afterwards, a list of extreme precipitation events associated with ARs was obtained by considering observations where the 12h-cumulated precipitation averaged over Portugal (using the surface stations) exceeded the 95th percentile , considering: i) only events with  spatially averaged precipitation >0.1mm; ii) that an AR was detected simultaneously in the region (IVT >450kg/m/s), following the threshold found by Ramos et al. (2015) for ERA-Interim reanalysis.*
*In addition, for the forecasts of extreme IVT and precipitation, we computed the 95th percentile of the corresponding period of analysis (2012-2016). To the computation of the percentile we had into account the data for the winters spanning between 2011-2012 and 2015-2016 and have defined the specific percentiles for each forecast day -1 to -14.*
*These forecasts are then compared against extreme precipitation observations, considering a "yes" forecast if a sufficient number of Ensemble members surpass that given threshold. The minimum fraction of ensemble members presenting a "yes forecast" varies between 0.1 and 1. So that 0.1 means 10% of the ensemble members have a "yes forecast", while 1 corresponds to the totality of the ensemble members. A ROC curve is then obtained by computing Hit Rates versus False Alarm Rates (Wilks, 2006), and considering these different minimum fraction of Ensemble members above the 95th percentile.*

In Fig 3 may be is not necessary adding all subpannels with all the days. Perhaps the authors can incorporate subpanels only every 3 days would be sufficient to show the idea and not overcharge the figure. These are very small and hard to visualize.

We agree with the reviewer that different forecast subpanels where very hard to read. This particular case was chosen to show the relatively larger differences that occur in the IVT field and intensity in the different lead times. Therefore, we choose to maximize the visible area of the sub-panels which allows us to keep the entire set of the forecast, and at the same time increasing the figure readability.

Fig 8 has too much information and of the different type. Considering splitting into two figures: the upper (percentages) and lower (contingency tables) panels. In caption indicate that the percentages correspond to the case study shown in Fig 3. The shading color codes of bars in contingency tables could be discrete (using the 5 subdivision) rather than continuous to better visualize the percentages. The first sentence in caption for these contingency tables could be rewritten as follow: Contingency tables for the accuracy of AR-related IVT forecasts by the ECMWF ensemble system, for lead times ranging between 1 and 15 days during winters spanning 2012-2016.

We agree with both reviewers comments regarding Figure 8. Therefore, it was divided into two separate figures, the new Figures 7 (percentages) and 8 (contingency tables). In addition, all the suggestions of improvement were included in the new version of the figure.

*Figure 7 (former upper panel of Fig. 8): Percentage of Ensemble members forecasting IVT above 450 Kg/m/s in each of the regional boxes and for each lead time for the case study presented in Figure 3 (January 4 2016). Green bars represent a spatially accurate forecast (in the box where the maximum IVT was observed). Yellow bars represent a forecast in an adjacent box to where it was actually observed. Red bars represent a forecast in one of the remainder boxes. The bars in the last line represent a completely missed forecast, by either: i) no AR forecast; ii) AR forecast outside of the 6 considered boxes in Western Iberia. (upper panel).*

*Figure 8 (former lower panel of Fig. 8): Contingency tables for the accuracy of AR-related IVT forecasts by the ECMWF ensemble system, for lead times ranging between 1 and 14 days, during the winters spanning 2012-2016. The red shading represents the percentage of observations versus forecasts. Note that a perfect forecast system would only present shadings in the diagonal, as the y-axis represents observed events in each box (as presented in Figure 2) and the x-axis represents forecasts in each box. The number of events in each box is shown in the y-axis by the blue arrow. The last row/column represent either: i) observations/forecasts outside of the 6 considered boxes; ii) no AR observed/predicted (lower panels)*

**Reviewer #2**

This manuscript quantifies the predictability and forecast skill of winter time atmospheric rivers affecting the Iberian peninsula using ensemble forecasts from ECMWF. Given the impact of precipitation associated with atmospheric rivers in society this is a very worthwhile study. The main results include that integrated water vapour transport is more skilfully predicted than precipitation at longer lead times and that the IFS has a systematic error which results in landfall of the atmospheric rivers being predicted too far north. While most of the manuscript is easy to understand, some parts such as the explanation of the diagnostics and what observations / analysis the forecasts are verified against are hard to understand and lacking critical details. These two major points are further explained in major comments 1-8 and other minor issues and typos which should be addressed are described under minor comments.

**Major comments:**
1. Section 2.1, lines 111 – 113. Here it is stated that daily values of IVT and precipitation from the IFS are used. Please clarify what is actually done here. I assume for precipitation it is the daily accumulated (so time integrated over 24 hours from 00 UTC - 00 UTC) precipitation but it is not clear what is meant by the daily IVT. Is this also integrated over time (24 hr) at each point?
For the IVT we considered instantaneous values, at 00UTC and 12UTC. These are compared with 12h-cumulated precipitation, centered at that time-steps. For example, 12UTC IVT is compared with precipitation falling between previous 06UTC and the following 18UTC. We acknowledge these details were not sufficiently clear in the original version of the manuscript, so we added this information.

The new version of the text reads as follows:

*The data considered here consists in instantaneous IVT values (for both direction and magnitude), at 00UTC and 12UTC, and 12-h accumulated precipitation centered in these time steps. The IVT was computed using the specific humidity and zonal and meridional winds between 300hPa and 1000hPa levels (e.g. Ramos et al., 2015).*

2. If IVT is integrated over time, does this act to smooth out (or zonally blur) the ARs and how does this impact the skill scores and the predictability. Previous studies for other forecast variables such as clouds and radiation (e.g. Hogan et al, Tuononen et al, 2019) have shown that while 24-hour integrated values are forecast with a large degree of skill, 6 hourly and 1 hourly values have much less skill.
As mentioned in the previous comment the IVT is not integrated over time, corresponding to instantaneous values, at 00UTC and 12UTC, so we don´t have the smoothing problem as the reviewer mentioned. Please see comment above.

3. Section 2.2, lines 124. Here it is stated that precipitation observations are accumulated into 12 hourly periods whereas the forecast precipitation is 24 hour accumulated values. Is this correct? Please clarify the time accumulations.

We agree with the reviewer that this information was not clear in the text. Both observation and forecast precipitation are accumulated into 12 hourly periods centered at 00UTC and 12UTC. Therefore, the 12UTC embraces the precipitation that occurs between 06UTC and 18 UTC for the same day, while the 00UTC embraces the precipitation registered between 18UTC from the previous day and the 6UTC of the following day.

*The 10 minutes precipitation were accumulated into consecutive 12h periods centered at 00UTC and 12UTC of each day. Therefore, the 12UTC embraces the precipitation that occurs between 06UTC and 18 UTC for the same day, while the 00UTC embraces the precipitation registered between 18UTC from the previous day and the 6UTC of the following day.*

4. Section 2.2. Were these precipitation observations, that are used to verify the forecasts, assimilated into these forecasts or are these independent observations?

The precipitation observations are only used in section 3. In addition, it must be noted that precipitation station data observations are not assimilated by the ECMWF Integrated Forecasting System, and therefore the observed dataset is totally independent from forecasts. The feasibility of assimilating SYNOP rain gauge data in the ECMWF model has been evaluated (see Lopez, P. Experimental 4D-Var Assimilation of SYNOP Rain Gauge Data at ECMWF. Mon. Weather Rev. 2013, 141, 1527–1544.) showing a very small impact in the operational system due to the large amount of other data already assimilated. The only rainfall product assimilated operationally by ECMWF is the Radar data over USA (Lopez, P. Direct 4D-Var Assimilation of NCEP Stage IV Radar and Gauge Precipitation Data at ECMWF. Mon. Weather Rev. 2011, 139, 2098–2116.).

5. It is hard to follow how the forecasts for IVT are verified. It is said later on in the manuscript that the analysis fields are taken from ECMWF to verify IVT but this should be mentioned much earlier, for example after section 2.2. This is because it is confusing to read how precipitation forecasts will be evaluated but not the IVT forecasts. I am also not sure if the precipitation analysis is used or not, and if not, why not.

Regarding the IVT, the forecast validation is entirely model based, using the analysis field from the ECMWF. We agree that this information was not clearly provided in the submitted version of the manuscript, especially the difference between IVT validation and precipitation validation. This information is now included in the new version of the manuscript.

Based on reviewer #1 comment and also on reviewer #2 comments the fist part of section 3, reads as follows:

*Firstly, a Receiver Operating Characteristic (ROC, Wilks, 2006) curves analysis was performed for IVT and precipitation forecasts for mainland Portugal. To begin with, using the observed precipitation dataset presented in Section 2.2, the mean precipitation (averaged over all mainland Portuguese stations) was computed. Afterwards, a list of extreme precipitation events associated with ARs was obtained by considering observations where the 12h-cumulated precipitation averaged over Portugal (using the surface stations) exceeded the 95th percentile , considering: i)*

*only events with spatially averaged precipitation >0.1mm; ii) that an AR was detected simultaneously in the region (IVT >450kg/m/s), following the threshold found by Ramos et al. (2015) for ERA-Interim reanalysis.*

*In addition, for the forecasts of extreme IVT and precipitation, we computed the 95th percentile of the corresponding period of analysis (2012-2016). To the computation of the percentile we had into account the data for the winters spanning between 2011-2012 and 2015-2016 and have defined the specific percentiles for each forecast day -1 to -14.*

*These forecasts are then compared against extreme precipitation observations, considering a "yes" forecast if a sufficient number of Ensemble members surpass that given threshold. The minimum fraction of ensemble members presenting a "yes forecast" varies between 0.1 and 1. So that 0.1 means 10% of the ensemble members have a "yes forecast", while 1 corresponds to the totality of the ensemble members. A ROC curve is then obtained by computing Hit Rates versus False Alarm Rates (Wilks, 2006), and considering these different minimum fraction of Ensemble members above the 95th percentile.*

6. Section 4. Line 155. How do you verify the precipitation over the boxes which are located over sea where there are no observation stations?

The precipitation forecast and validation is only done in section 3, where the authors compare the predictive skill of precipitation and IVT. Based on the results obtained in section 3 (higher IVT predictability), all the remaining results sections (i.e. sections 4 and 5) are focused specifically on the IVT (ARs) predictability.

Taking this into account we now stress, at the end of section 3, that only the IVT will be analyzed from this point onwards:

*Based on the results presented in Figure 1, we show that the IVT can provide an added value for mid-range operational forecast of extreme precipitation events. Therefore, from this point onward we will focus our analysis on the performance of the ECMWF probabilistic forecasts for IVT and the AR-related IVT forecasts over Portugal, exploring potential systematic biases, and trying to access model behavior and accuracy metrics at different forecast lead times.*

7. Section 4.1, lines 165 – 171. It is very hard to understand these diagnostics and as such this is the biggest weakness of this manuscript. This must be improved. Specific points are:

(a) Landfall distance. As this is described (line 165) this is the scalar distance simply measured between two points which in theory should always have a positive value and no direction. However when this is discussed in the text and shown in Figure 4 this parameter can have negative values and a direction. Is this then the difference in the meridional direction with positive (negative) errors indicating a northward (southward) forecast relative to the analysis? Please clarify.

We agree with the reviewer that the wording used in the submitted version of the manuscript lacked sufficient clarity. Regarding the landfall distance, the values presented correspond simply to the meridional distance (in km) between the landfall (location of the maximum IVT) in the analysis and in the forecast. As stated by the reviewer, this value can be positive (negative) errors indicating a northward (southward) forecast landfall error.

(b) How is the landfall location identified? Is this the first point in time when IVT exceeds the threshold value over a land point in any of the boxes? Again please clarify this in the revised manuscript.

The landfall location corresponds to the latitude of the maximum IVT within the coastal area used for detection (the box domains presented in figure 2).

(c) The landfall IVT error is sensitive to both intensity and displacement errors. This should be noted more clearly. It would also be interesting to include a diagnostic which solely measures the intensity error e.g. the difference in the maximum value in the forecast and the analysis regardless of where they occur.

We agree with the reviewer that the landfall IVT error is sensitive to both intensity and displacement errors. That is why we developed three different metrics to test it: (1) Landfall distance and (2) Landfall IVT error and (3) AR-axis IVT error.

Regarding the suggestion raised by the reviewer to have a new diagnostic that measures the intensity error, this is already included in metric (3), which is exactly the difference in the maximum value in the forecast and in the analysis, regardless of where these two maxima occur. We believe that with the new addition to the text, this information will become clear.

(d) The AR-axis angle error. Two points (or a vector) are always need to calculate an angle e.g. you need to identify the axis of the AR yet this is not done here. I do not fully understand how this angle is calculated in the forecast / analysis and therefore I do not understand how the difference can be calculated. I assume it is the angle of the IVT vector but where and when? Please clarify this. A schematic diagram may be helpful as would adding the IVT vectors to the large panel in Figure 3 to make it clearer to readers that IVT is vector and the shading is the magnitude of that vector.

The AR-axis angle is relative to the landfall region, not to the "entire" AR, in this regards it is more appropriate to state that we compute the angle of incidence of the AR in the target area. In this sense, it is not very easy to depict it as suggested in Fig.3, due to the relatively small spatial scale. Regarding its computation, we simply detect the latitude of the maximum IVT for each longitude within the target area. Then, using those latitudes, the "mean" angle is calculated, using a west-east direction as the 0º reference. As for other metrics, this is computed for analysis and forecast, providing the error in the angle. Positive (negative) errors denote a counterclockwise (clockwise) error. We added this information in the revised manuscript, to make it clearer.

Considering the review comments the diagnosis description now reads as follows:

*Afterwards, forecasts up to 14 days in advance from the control and ensemble members where compared against the analyses, through the computation of the following metrics that consider the landfall IVT error sensitivity to both intensity and displacement errors:*

*1)      Landfall distance: the meridional distance (in km) between the landfall (location of the maximum IVT) in the forecast and in the analysis. This value can be positive (negative), indicating a northward (southward) forecast landfall error.*

*2)	Landfall IVT error: the difference (forecast minus analysis) between the IVT (in kg/m/s) at the correct location of the landfall, i.e., where the maximum IVT was actually observed in the analysis;*

*3)	AR-axis IVT error: the difference (forecast minus analysis) between the IVT (in kg/m/s) at the specific individual locations of the landfall in the analysis and forecast. It considers the difference in the maximum IVT value in the forecast and the analysis, regardless of where they occur;*

*4)	AR-axis angle error: the difference (forecast minus analysis) between the incidence angle (in º, respective to W→E) at the specific locations of the landfall (Figure 2) in the analysis and forecast. The latitude of the maximum IVT is detected for each longitude within the target area. Then, using those latitudes, the "mean" angle is computed, using a west-east direction as the 0º reference. As for other metrics, this is computed for analysis and forecasts, providing the error in the angle. Positive (negative) errors denote a counterclockwise (clockwise) error.*

8. It is not clear how the diagnostics described in section 4.1 are calculated in the cases that no AR is forecast. Are these included as missing data? How does this impact the overall results and conclusions? Please add some information about this.

To create a catalogue of Observed events we only considered analysis where the 450kg/m/s threshold is surpassed within the target area (boxes). However, the maximum IVT (intensity and location) is detected in a much wider latitudinal window (further north/south). For example, in the Case Study presented to explain the methodology (new Figure 4), if the maximum IVT is detected further north/south than the target domain, or if the maximum is below 450kg/m/s, it is considered as no-AR in the domain (as depicted by the open circles). However, regardless of being detected as AR in the domain or not, the maximum IVT at the longitudes where the target area is located (as well as the latitude where that maximum is located) is always kept. These values are used for the computation of "mean statistics" presented in figure 5 and figure 6 (new version of the manuscript), in the same way that values Forecasted as AR are.

**Minor comments and typos:**
1. Title. I'm not 100% sure this title is grammatically correct. Would "Predictive skill of atmospheric rivers in the western Iberian Peninsula" be more correct?
The title was changed.

2. Line 75. Should read "These kind of studies..."
The sentence was corrected.

3. Lines 77-80. The information presented here about the AR reconnaissance program is somewhat out of place. Either this program needs to be further explain and links made to the research presented in this paper or this should be removed.
We agree with the reviewer that the AR reconnaissance program sentence was not needed in the context of this paper. Therefore, it was deleted from the text.

4. Line 91. What is meant by this statement "The EFI for IVT became control at ECMWF…."? Please clarify the text here. I think it should read "became operational at..."
Thank you for spotting this error. The information was corrected in the text.

5. Line 98 / objective 1. This objective does not make sense. I think what it meant here is to compare the impact of forecast lead time of the forecast values of both IVT and precipitation. Please revise.
The objective 1 was revised in order to become clearer.: "*The main objective here is twofold: a) the comparison between the predictive skill of precipitation and IVT at different lead times during extreme ARs striking western Iberia, using ECMWF ensemble forecasts up to 15 days for winters between 2012/2013 and 2015/16;….. *"

6. It would be helpful to add letters to the panels in the figures and refer to the panels using the letters rather than "upper panel" etc.
We agree with the reviewer's suggestion. Therefore, letters were added to all the figures with multiple panels. In addition, the text was also changed accordingly.

7. Line 290. There is a typo in the reference here and "to" is missing in the sentence "This is due the…."
Thank you for spotting that. The typo was corrected.

8. Line 309. "control context". I think what is meant here is "in an operational context...".
The text was corrected.

**Figure comments:**
1. Figure 1, top panel. The colour bar is hard to read since it is an continuum. Can this be changed to have discrete colours and only the number of colours that there are lines on this figure (I think 4 colours). The yellow lines are also hard to see so using darker colours would be better.
We agree with the reviewer, and changed the figure accordingly. Also, confidence intervals have been added to panel b), regarding the "area under the ROC curve", following a bootstrap procedure.

2. Figure 1. bottom panel. What is the grey bar above this panel for?
This was a problem with the figure exporting the title for the panel. We thank the reviewer for noticing it, and it is corrected in the revised version of the manuscript.

3. Figure 3. The boxes are hard to see in the top panel as they are similar colours to the shading. Furthermore, the panels at the bottom are very small and hard the see. These smaller panels would be clearer if the area boxes were removed and if the titles were shortened as this would allow the images to be made larger.
We agree with the reviewer that different forecast subpanels where very hard to read. This particular case was chosen to show the relatively larger differences that occur in the IVT field and intensity in the different lead times. Therefore, we choose to maximize the visible area of the sub-panels, which allowed us to keep the entire set of forecasts, and to increase the figure readability.

4. Figure 4 caption. "Solid blue line represents the error in the location of the maximum IVT between observations and each forecast". What observations of IVT is available or should this read "...between the verifying analysis and each forecast". Also see major point 5 above.

We agree with the reviewer that this is not clear in the text. As mentioned in reply to major point 5, the error is between the verifying analysis and each forecast. As mentioned before, figures 4 and 5 were combined in this new Figure 4 and the caption was changed accordingly.

*Figure 4. Example of the evolution with lead time for the accuracy of IVT probabilistic forecasts, for the event presented in Figure 3. In a) the black line represents the error in the location of the maximum IVT (i.e. landfall distance) in the Operational run (in km), while the blue thick solid line represents the landfall distance for the Ensemble Forecasts. The blue shaded envelope accommodates the Ensemble spread, considering the 25th and 75th percentiles. In addition, the black arrows represent the errors in the angle (in degrees) of the AR axis for each forecast. Panel b) shows the error in the IVT intensity (Kg/m/s) for each forecast at the observed landfall location. Black solid line, red solid line and red shaded envelope are as in panel a). Panel c) shows the error in the maximum IVT at the specific locations where it has been observed and forecasted for each lead time, regardless of the landfall distance. Black solid line, dashed red line and red shaded envelop as in a) and b). The open circles represented in some lead times represent forecasts where the maximum IVT did not surpass a minimum threshold of 450 Kg/m/s within the target domain (i.e. regional boxes over Western Iberia).*

5. Figure 5. The shading is not very clear in the top panel and appears to change shade. Can this be improved? Also please add information to the caption about how the "spread" is calculated. For example, is this the maximum and minimum differences or the 25th - 75th percentile that is shaded?

Figure shading in this figure (old Figure 5, part of new Figure 4) was improved in order to become clearer. The information about the percentiles has been added to the caption.
See previous comment.

6. Figure 8 is very small and hard to see. The caption is also very long and hard to follow. Could this figure be split into two figures e.g. top panel and then the middle and bottom panels as a separate figure?

As suggested by both reviewers, we split Figure 8 into two figures (New figure 7 and 8), and the captions where changed accordingly.

7. Figure 9 is also hard to see and could be made large. The colours could be explained briefly in the caption here rather than expecting a reader to return to Figure 2. e.g. The darkest blue bar represent the most northerly box and the yellow bars the most southerly box.

We agree with the reviewer suggestion. The figure was improved in order to become clearer and the caption was also improved. It reads as follows:

*Figure 9. Forecast verification metrics for IVT exceedances (>450 Kg/m/s) using the ECMWF Ensemble forecast system during the 2012-2016 extended winters in Western Iberia, and for lead times between 1 and 14 days. Colored bars represent metrics for individual regional boxes, as*

*where the darkest blue bar represents the most northerly box and the yellow bars the most southerly box (as depicted in Figure 2).*

The relevance of these metrics can be easily understood looking at a case study. In Fig.3 ((a), analysis), an example of an AR affecting the North Portugal box is presented. The 14 small panels (Fig. 3b) show how the control forecast changed with increasing forecast daily lead time. While at short lead times the location, intensity and angle are quite similar to the analysis, at longer lead times the control forecast becomes less accurate, and some of them predicting an IVT magnitude, axis orientation and landfall totally disconnected from reality. This example higliths the importance of considering ensemble forecasts for long lead times, as discussed below.

The evolution of the forecasts is depicted in Fig. 4, where the metrics for each lead time are summarized using both control and ensemble. As lead time increases, it is notable how the AR was being predicted further north by the control forecast (Fig 4a) , black line). In fact, for lead times of -9 days, -11 days and -14 days, there was no predictive skill of an AR affecting the defined boxes (either forecasted much further north, or not forecasted at all, as depicted by the open circles), and in accordance with the corresponding subplot observed in Fig. 3. Consequently, as the landfall distance error increases, the IVT error at that location also increases (black line line in Fig. 4b). When considering the AR-axis IVT error (Fig. 4c, dashed black line), the decrease with longer lead times is smaller, showing that despite the error in the actual position of the AR, its maximum intensity was well forecasted for most of the period. Regarding the angle of incidence of the AR, it was mostly zonal, both in the analysis and most forecasts, thus with small error. Nevertheless, and as seen in Fig. 3, during mid-range lead times (around 7-10 days) forecasts tended to tilt the AR NW→SE over Western Iberia, as depicted by the arrows in Fig. 4a).

Moreover, we computed the same metrics described above for each of the 50 ensemble members of ENS, being these results are also presented in Fig. 4. The errors of the ensemble mean are presented for landfall distances (blue thick line) and Landfall IVT error (red thick line) and AR-axis IVT error (dashed black line)

as well as the spread in the ensemble forecast (shaded areas), shown here as the 25th and 75th percentiles of the ensemble computed metrics distribution. The control forecast errors and the ensemble mean are in good agreement with the dispersion of the ensemble forecast increasing with lead time. Regarding the landfall distance, it was found that the error increases with lead time, and in this case the control forecast error at lead time -12 days is higher than the ensemble mean and even the ensemble dispersion. The 50 members of the ensemble for lead time -5 day (Fig. S1) and for -12 days (Fig. S2) are shown in supplementary material Fig. S1 where it can be compared with the control IVT forecast shown in Fig. 3. One can see, that the ensemble members for the shorter lead time are in better agreement and closer to reality. When looking to -12 days lead time (Fig. S2), the dispersion of AR location is much higher when compared with the reality and even with the control forecast shown in Fig. 3. In addition, both Landfall IVT error and AR-axis error (Fig. 4), for both control and ensemble members, show a decrease in the forecasted IVT as we consider longer lead times.

**4.2. Mean performance of the ECMWF forecasts during 2012-2016**

In the previous section we have analyzed one case study, evaluating both control and ensnemble forecasts against analysis. We extended the evaluation of the IVT forecast metrics for all events occurring during the extended winters of the study period (from 2012-2013 to 2015-2016). Similarly to the case study presented in Fig. 4., the same metrics have been computed for all AR landfall cases (207), and the average error is obtained as presented for the control forecast in Fig.5 and for the ensemble forecast in Fig. 6.

When analyzing the control forecast of ARs over Western Iberia (Fig .5), it is possible to identifysome systematic errors/biases. Regarding the mean errors (Fig. 5a) a northward landfall bias is systematically found for longer lead times, especially for those longer than 5 days, which can reach up to 800km at +14 days. In addition, regarding the AR-axis angle error, the results show a slightly negative bias in respect to those actually observed. Since we consider the 0º angle as west-east orientated and that AR events over Portugal tend to present a southwest-northeast orientation (Ramos et al., 2015), this systematic bias reflects a lower tilt in the AR orientation at longer time forecasts, or in other words, a tendency for more zonal forecasts (Fig. 5a). When analyzing the IVT magnitude, both landfall IVT error and AR-axis IVT error have a negative bias, as a result of: i) the error in the landfall location; ii) an underestimation of the ARs intensity. Comparing both metrics in more detail, it can be noted that the ARs-axis IVT bias is lower than the landfall IVT error, showing that while the IFS is forecasting the intensity of the ARs quite well (with just a small underestimation in intensity associated to lead times), the landfall location bias leads to significant IVT forecast errors at the location where the AR actual landfall is observed. The mean absolute errors were also computed for the same metrics (Fig. 5b), unsurprisingly presenting higher errors for landfall distance, location biases occur both northwards and southwards, thus partially canceling themselves, as shown in Fig. 5a. Nevertheless, this difference is not that large, thus reinforcing the systematic tendency for a bias towards the north in longer lead time forecasts. A similar rationale is applicable to the incidence angle, where errors in the tilt of the ARs in different directions tend to cancel out. As so, mean absolute errors tend to be around 45º at longer lead times (Fig. 5b). Overall it is possible to affirm that the case study evaluated in Fig. 4 presents similar biases to those obtain here with the average of the entire set of ARs considered.

The mean weighted errors/biases of the ensemble forecasts (i.e. all the 50 ensemble members) for all events are presented in Fig. 6. The same methodology as in Fig. 4 is followed here, and the ensemble

mean is presented, along with the spread in the ensemble forecast (shown here as the 25th and 75th percentiles forecast distribution). The results are very similar to the ones found for the control forecast (Fig. 5) with a positive bias (northly) in the position of the AR landfall as we move to higher lead times, along with a negative bias (less AR intensity) in the landfall IVT error and AR -axis IVT error. However, the dispersion in the ensemble forecast is higher in the landfall distance than the in the other IVT error metrics, reinforcing that the model forecasts the ARs but lacks skill in forecasting the right location of their landfalls.

It is vital to stress the use of the ensemble forecast in this kind contitions. In Fig. 4 and Fig. 6., we compare the control forecast with the ensemble mean and the ensemble errors metrics. Both control forecast and ensemble forecast Landfall and IVT error shows a northly bias and a negative IVT bias on the landfall location, respectively. Ramos et al., 2016, showed that the number of ARs affecting Iberia is relatively lower when compared with the contiguous western France or even the UK using the ERA-Interim reanalysis. This means that on average most of the AR go further north, and the ones hitting Iberia are not that frequent. Taking this into account, one can hypothesize, that the northerly bias and respective negative bias in IVT intensity in the ARs forecasts at longer lead times can reflect that the model tends to its climatology, which is to have ARs 
[revised manuscript text omitted]

This work was supported by the project Landslide Early Warning soft technology prototype to improve community resilience and adaptation to environmental change (BeSafeSlide) funded by Fundação para a Ciência e a Tecnologia, Portugal (FCT, PTDC/GES-AMB/30052/2017). P.M.S. and R.M.T. were supported by the project Improving Drought and Flood Early Warning, Forecasting and Mitigation using realtime hydroclimatic indicators (IMDROFLOOD) funded by FCT (WaterJPI/0004/2014). A.M.R. was also supported by the Scientific Employment Stimulus 2017 from FCT (CEECIND/00027/2017). The authors would like to thank David Lavers, from the ECMWF, for the helpful comments on this manuscript.

**References**

[revised manuscript text omitted]

Figure 8. Contingency tables for the accuracy of AR-related IVT forecasts by the ECMWF ensemble system, for lead times ranging between 1 and 14 days, during the winters spanning 2012-2016. The red shading represents the percentage of observations versus forecasts. Note that a perfect forecast system would only present shadings in the diagonal, as the y-axis represents observed events in each box (as presented in Fig.2) and the x-axis represents forecasts in each box. The number of events in each box is shown in the y-axis by the blue arrow. The last row/column represent either: i) observations/forecasts outside of the 6 considered boxes; ii) no AR observed/predicted.

Figure 9. Forecast verification metrics for IVT exceedances (>450 Kg/m/s) using the ECMWF Ensemble forecast system during the 2012-2016 extended winters in Western Iberia, and for lead times between 1 and 14 days. Colored bars represent metrics for individual regional boxes, as where the darkest blue bar represents the most northerly box and the yellow bars the most southerly box (as depicted in Fig. 2).